# The Impact of p70S6 Kinase-Dependent Phosphorylation of Gemin2 in UsnRNP Biogenesis

**DOI:** 10.3390/ijms242115552

**Published:** 2023-10-25

**Authors:** Lea Marie Esser, Qiaoping Li, Maximilian Jüdt, Thilo Kähne, Björn Stork, Matthias Grimmler, Sebastian Wesselborg, Christoph Peter

**Affiliations:** 1Institute of Molecular Medicine I, Medical Faculty, Heinrich Heine University Düsseldorf, 40225 Düsseldorf, Germany; 2Institute of Experimental Internal Medicine, Otto von Guericke University, 39120 Magdeburg, Germany; 3Institute for Biomolecular Research, Hochschule Fresenius gGmbH, University of Applied Sciences, 65510 Idstein, Germany; 4DiaServe Laboratories GmbH, 82393 Iffeldorf, Germany

**Keywords:** UsnRNP biogenesis, Gemin2, p70S6K, mTOR pathway, post-translational modifications

## Abstract

The survival motor neuron (SMN) complex is a multi-megadalton complex involved in post-transcriptional gene expression in eukaryotes via promotion of the biogenesis of uridine-rich small nuclear ribonucleoproteins (UsnRNPs). The functional center of the complex is formed from the SMN/Gemin2 subunit. By binding the pentameric ring made up of the Sm proteins SmD1/D2/E/F/G and allowing for their transfer to a uridine-rich short nuclear RNA (UsnRNA), the Gemin2 protein in particular is crucial for the selectivity of the Sm core assembly. It is well established that post-translational modifications control UsnRNP biogenesis. In our work presented here, we emphasize the crucial role of Gemin2, showing that the phospho-status of Gemin2 influences the capacity of the SMN complex to condense in Cajal bodies (CBs) in vivo. Additionally, we define Gemin2 as a novel and particular binding partner and phosphorylation substrate of the mTOR pathway kinase ribosomal protein S6 kinase beta-1 (p70S6K). Experiments using size exclusion chromatography further demonstrated that the Gemin2 protein functions as a connecting element between the 6S complex and the SMN complex. As a result, p70S6K knockdown lowered the number of CBs, which in turn inhibited in vivo UsnRNP synthesis. In summary, these findings reveal a unique regulatory mechanism of UsnRNP biogenesis.

## 1. Introduction

The biogenesis of uridine-rich small nuclear ribonucleoproteins (UsnRNPs) plays a key role in post-transcriptional gene expression in eukaryotes, including pre-messenger RNA (pre-mRNA) splicing and the inhibition of premature termination [1,2,3]. Each UsnRNP is composed of a specific, uridine-rich small nuclear RNA (UsnRNA), as well as the seven Sm proteins B, D1, D2, D3, E, F, and G. The Sm proteins form a stable heptameric ring structure, binding the so-called Sm site (Sm core) on the common snRNAs U1, U2, U4 or U5 [4,5,6,7,8]. The assembly of the Sm core occurs stepwise in cells and is highly regulated by two cooperating protein complexes: the protein arginine methyltransferase 5 (PRMT5) and the survival of motor neuron (SMN) complex [9,10,11].

The SMN complex consists of the survival motor neuron (SMN) protein, Gemins 2–8, and the UNR interacting protein (UNRIP) [9,12,13,14,15,16]. Within the SMN complex, the SMN protein and Gemin2 act as a functional core, responsible for Sm protein binding, which forces the transfer of the Sm proteins onto the Sm site of the snRNA and thus the formation of a core snRNP [17,18,19]. After hypermethylation of the snRNA cap, the entire complex is reimported into the nucleus [19,20]. Before being active in mRNA splicing, the snRNPs mature in aggregates within the nucleus, the so-called Cajal bodies (CBs), containing SMN complex components and coilin [21,22,23].

Further research already revealed a crucial role for post-translational modifications as a key regulation regulator in the biogenesis of snRNPs [24,25,26,27,28]. Although several studies have showed that the SMN complex is highly phosphorylated, and the ATP-dependency of the snRNP core formation has been known for many years [11,26,27], the structural or mechanistic consequences of these post-translational modifications have not been identified so far. The work from Schilling et al. [29] recently identified kinases from the mTOR pathway as interacting with core components of the SMN complex. Our work presents here the important role of the Gemin2 protein within this complex. Further research [18] has identified Gemin2 as binding the pentamer of Sm proteins independent from pICln. These results are in contradiction with previous work [24,25,30,31], indicating that the assembly chaperone protein pICln is necessary for the binding of the Sm proteins. In our recent work, we were able not only to demonstrate the necessity of pICln concerning the binding of the Sm proteins but also to show the importance of pICln phosphorylation, catalyzed by the autophagy-activating Unc-51-like kinase (ULK1), another kinase from the mTOR pathway, which mediates the release of Sm proteins onto the SMN complex [28]. In this work, we identified Gemin2 as a new p70S6 kinase-specific binding partner and phosphorylation substrate. We focus on the characterization of the phospho-status of Gemin2 and its influence on nuclear SMN condensation, visualized via the number of nuclear CBs via immunofluorescence, as a readout for the efficiency of UsnRNP biogenesis. Furthermore, we demonstrate that the upstream phosphorylation of the mTOR-related kinase p70S6 regulates Gemin2 activity and, as a consequence, UsnRNP formation in vivo (Figure 1).

## 2. Results

### 2.1. The p70S6 Kinase Is a New Interaction Partner of the SMN Complex

The SMN complex is a multi-megadalton protein complex consisting of the SMN protein itself, Gemins 2–8, and UNRIP [9,12,13,14,15,16,32,33]. Gemin2 is, along with the SMN protein, one of the essential elements of the SMN complex [34]. It is well established that phosphorylation events play a major role in controlling the SMN complex, and Gemin2 may have putative phosphorylation sites [27]. Earlier studies have shown that mTOR/-p70S6K-dependent phosphorylation events play an important role in this context, although the molecular mechanisms are yet unknown [29]. Our present study aims to clarify the role of p70S6K as a regulator of UsnRNP formation (Figure 1). Immunofluorescence studies were conducted to ascertain the role of p70S6K. In line with previous work [9,35], Gemin2 localizes predominantly in the cytoplasm, as well as in so-called Cajal bodies (CBs) within the nucleus of the cell (Figure 2A). p70S6K also localizes in the cytoplasm (Figure 2B), where it co-localizes with Gemin2 (Figure 2C).

To test whether Gemin2 directly interacts with p70S6K, immunopurification studies were performed. To this end, we established an inducible expression system for the GFP-immunopurification of a GFP-Vector control (GFP), GFP-Gemin2 wt, and GFP-SMN wt in Flp-In T-REx 293 cells. In this context, it is interesting to note that each Gemin (Gemin2–5) and the SMN protein bind in amounts comparable to both GFP-SMN and GFP-Gemin2 wild type, indicating that the overexpressed proteins are functional in the formation of the entire SMN complex. However, only Gemin2 showed an interaction with p70S6K. Thus, we identified p70S6K as a new, Gemin2-specific binding partner of the SMN complex (Figure 2D). To further validate this survey, we performed endogenous immunopurification and immunoblot analysis from the two potential interaction partners Gemin2 and p70S6K, showing that the two proteins also interact in vivo (Figure 2E).

### 2.2. The SMN Complex Core Subunit Gemin2 Is a New Substrate of the p70S6 Kinase

To determine whether Gemin2 is a new and so-far unknown substrate of p70S6K, we performed in vitro kinase assays using recombinant purified GST-fusion proteins of Gemin2 and SMN as substrates and active 6xHis-p70S6K purified from Sf21 insect cells. All used substrates showed ^32^P incorporation upon incubation with active His-p70S6K. Although Gemin2 does not have many putative phosphorylation sites (two threonines and four serines) [27,29], it showed a strong ^32^P incorporation (Figure 3A). Based on earlier mass spectrometry results from Schilling et al. [29], the potential phosphorylation sites serine 81 and serine 166 within Gemin2 were mutated. Each potential phosphorylation site was mutated to an alanine, as a phospho-deficient, and to an aspartate, as a putative phospho-mimicking mutant. To verify these potential p70S6K phosphorylation sites within Gemin2, we performed in vitro kinase assays with GST-Gemin2 wt, GST-Gemin2 S81A, GST-Gemin2 S166A, GST-Gemin2 S81D, GST-Gemin2 S166D, and GST only. Gemin2 wt and Gemin2 S166’s mutation to alanine and to aspartate showed a strong ^32^P incorporation. Interestingly, the Gemin2 S81 mutation to alanine and to aspartate showed a strong decrease in phosphorylation signal, meaning that this was a p70S6K-specific residue (Figure 3B). However, the phosphorylation was not blocked completely, indicating that this was not the only p70S6K-specific phosphorylation site within Gemin2 in this setting. A pulldown assay was performed with these recombinant Gemin2 phosphorylation mutants to see whether the binding capacity of these proteins was also affected. GST-Gemin2 wild type (wt), GST-Gemin2 S81A, GST-Gemin2 S166A, GST-Gemin2 S81D, and GST-Gemin2 S166D bind SMN and SmB/B’ (Y12 antibody) to the same extent. This indicates that the mutations at the respective serines within Gemin2 do not affect the overall protein structure or formation of the SMN complex. Only the Gemin2 S81 alanine mutant showed a decrease in p70S6K binding. As this is also the mutant that showed a decrease in phosphorylation, these results indicate on one hand that S81 is the major site of phosphorylation but also may represent the area of direct binding of Gemin2 and p70S6K (Figure 3C).

### 2.3. The p70S6 Kinase-Dependent Phosphorylation Influences Binding towards Gemin2

We identified Gemin2 as a new, p70S6K-specific interaction partner and substrate; however, the question of the cellular effect of Gemin2 phosphorylation remained unclear. To further investigate the effect of the Gemin2 phospho-status, Flp-In T-REx 293 cells overexpressing GFP-Vector control, GFP-SMN wt, GFP-Gemin2 wt, GFP-Gemin2 A81, GFP-Gemin2 A166, GFP-Gemin2 D81, and GFP-Gemin2 D166 were used. To test the expression efficiency of the cell lines, immunoblot analyses were performed with cytoplasmic extract (S100) as well as a GFP-immunopurification of these extracts. All generated cell lines expressed the GFP-tagged fusion proteins in comparable amounts (Figure 4A). The localization of the GFP-tagged proteins was visualized using immunofluorescence analysis. While GFP-Vector control, GFP-Gemin2 wt, GFP-Gemin2 A81, GFP-Gemin2 A166, GFP-Gemin2 D81, and GFP-Gemin2 D166 localized in the cytoplasm and the nucleus, GFP-SMN wt mainly aggregated in dots in the cytoplasm of the cell (Figure 4B). To see whether the overexpression of GFP-Vector control, GFP-Gemin2 wt, and the GFP-Gemin2 phospho-mutants influenced endogenous Gemin2 complex building, S100 extracts of these cell lines were separated with size exclusion chromatography with a Superose 6 column. Via immunoblot analysis, when a pan-specific antibody against Gemin2 was used, it was evident that Gemin2 is distributed in a complex in the higher molecular size range around 2000 kDa and in addition in the distinct lower molecular weight range from 158 kDa to 100 kDa, comigrating with the so-called 6S complex, an RNA-free intermediate consisting of the chaperone protein pICln and the Sm proteins D1, D2, E, F, and G [36].

The overexpression of neither GFP-Gemin2 wt nor the GFP-Gemin2 phosphorylation-mutant variants of Gemin2 affected the Gemin2 distribution pattern (Figure 4C). To test the scenario of whether Gemin2 phosphorylation status influences the binding of interaction partners, a GFP-immunopurification with cytoplasmic extracts of those cell lines was performed. All core components of the SMN complex, Gemin2–5, and SMN bind in equal amounts to the Gemin2 phospho-mutants. This indicated that the complex composition of the SMN core-complex is not affected by overexpression (Figure 4D). 

### 2.4. The Phosphorylation Status of Gemin2 Influences UsnRNP Biogenesis and Inhibition of p70S6K Results in a Decreased Number of Cajal Bodies

So far, this study identified p70S6K as a specific interaction partner of the SMN complex and revealed Gemin2 as a phosphorylation substrate of this kinase. This data raised the question of whether the Gemin2 phospho-status has a direct impact on the regulation of UsnRNP biogenesis. In the late stage of UsnRNP biogenesis, the UsnRNP matures in an aggregate in the nucleus, within the CBs, which can be visualized as SMN and coilin positive dots via immunofluorescence [21,28]. To test the SMN condensation, the GFP-Vector control, GFP-SMN wt, GFP-Gemin2 wt, GFP-Gemin2 A81, GFP-Gemin2 A166, GFP-Gemin2 D81, and GFP-Gemin2 D166 overexpressing cells were seeded on coverslips and stained with antibodies against SMN and coilin (Figure 5A). All numbers of CBs mentioned in this work are the mean of 1000 cells per condition. The overexpression of GFP-SMN wt (2.44) caused a strong increase in the number of CBs compared to the GFP-Vector control (1.19). Expression of GFP-Gemin2 wild-type protein and the corresponding phospho-mutants had only a slight effect on the number of Cajal bodies (Figure 5B); this can be explained by the presence of endogenous Gemin2. Therefore, we treated HEK293T cells with p70S6K-siRNA, followed by immunofluorescence against SMN and coilin, to visualize the quantity of CBs (Figure 5C,D). The number of CBs per nucleus (mean) was significantly decreased in p70S6K knockdown HEK293T cells (0.75) compared to untreated control HEK293T cells (1.43) and cells transfected with non-target control siRNA (1.31) (Figure 5D). 

Taken together, Gemin2 is a newly identified p70S6 kinase substrate. The mutation of Gemin2 phospho-sites has an impact on the binding of the kinase, but not on SMN complex integrity. Knockdown of p70S6K in cells has also a strong impact on UsnRNP biogenesis, indicating that this kinase influences the regulation of this process via phosphorylation of Gemin2. 

## 3. Discussion

The SMN complex fulfills a key role in the biogenesis of uridine-rich small nuclear ribonucleoproteins (UsnRNPs), which is a well-organized stepwise mechanism highly regulated via post-translational modifications [24,25,26,27]. Recent research has demonstrated a significant role of kinases from the mTOR pathway in this context [28,29,36]. In our study presented here, we show that p70S6 kinase interacts with the SMN complex via Gemin2 and influences UsnRNP biogenesis via phosphorylation of Gemin2 (Figure 1). 

Immunofluorescence analyses showed colocalization of Gemin2 and p70S6K in the cytoplasm (Figure 2A–C). Subsequent endogenous immunopurification studies have revealed that p70S6K is co-precipitated with Gemin2 but also interacts with it via direct binding (Figure 2D,E). The significantly weaker detection of p70S6K in the GFP-SMN IP compared with the GFP-Gemin2 IP likely indicates indirect binding via endogenous coprecipitated Gemin2 (Figure 2D).

This is consistent with previous work showing that phosphorylation of the SMN complex and the resulting regulation occurs predominantly in the cytoplasm [26,28]. Although in vitro phosphorylation studies have demonstrated that both SMN and Gemin2 proteins can be phosphorylated via p70S6K (Figure 3A,B), results from immunopurification studies suggest that Gemin2 is the major substrate and binding partner of p70S6K within the SMN complex (Figure 2D,E).

The phosphorylation sites of Gemin2, serine 81 and serine 166, as postulated in the work of Schilling and colleagues, were mutated to an alanine, as a phospho-deficient, and to an aspartate, as a phospho-mimicking mutant [29]. Interestingly, in in vitro kinase assay, Gemin2 S81 mutations to alanine and to aspartate showed a strong decrease in phosphorylation signal, identifying serine 81 as a p70S6K-specific target (Figure 3B). Nevertheless, the phosphorylation was not blocked completely, meaning there may be more phosphorylation sites within Gemin2, consistent with the earlier mass spectrometry analysis by Schilling and colleagues [29]. The in vitro binding studies using these recombinant Gemin2 phosphorylation mutants revealed that the binding towards SMN and SmB/B’ is not affected, indicating that the phospho-mutations do not affect SMN core complex building (Figure 3C and Figure 4D). Indeed, the Gemin2 S81-to-alanine mutant showed a decrease in p70S6K-dependent phosphorylation and a slight decrease in p70S6K binding, suggesting that the reduced binding affinity of the Gemin2 S81 mutant affects the phosphorylation efficiency, too (Figure 3B,C).

To address the question of the intracellular effect of Gemin2 phosphorylation, cells which inducibly overexpress GFP, GFP-SMN wt, GFP-Gemin2 wt, GFP-Gemin2 A81, GFP-Gemin2 A166, GFP-Gemin2 D81, and GFP-Gemin2 D166 were generated (Figure 4A,B). Size exclusion chromatography pointed out that Gemin2 is distributed in a higher molecular complex as well as in the distinct molecular weight range from 158 kDa to 100 kDa, the same size range as the 6S complex [36]. This suggests that the 6S complex, consisting of pICln and the Sm proteins D1, D2, E, F, and G, might be a kinetic trap involved in the transfer of Sm proteins onto the SMN complex to keep snRNP biogenesis ongoing [24]. Our findings now identify Gemin2 as co-migrating in the size of the 6S complex by using size exclusion chromatography in vivo (Figure 4C). Further in vitro studies revealed Gemin2 to organize the stepwise formation of the Sm core by binding the Sm protein pentamer and at the same time prevent it from binding RNAs. Gemin2 was shown to have an open confirmation, where the C- and N-terminus of the protein wrap around the pentamer of the Sm proteins SmD1/D2 and SmE/F and G, connected by an unstructured loop region [18]. This data strengthens the assumption that Gemin2 is a bridging factor between the 6S complex and the SMN complex involved in the transfer of the Sm proteins from the 6S complex onto the snRNA [18,37]. 

During UsnRNP biogenesis, the snRNPs mature in the nucleus, within the CBs [21,28]. According to earlier studies [27,29], the subcellular location of the SMN complex and, consequently, its nuclear condensation in CBs, is controlled via the phosphorylation of serine/threonine residues as a regulatory mechanism. Due to this, we investigated how Gemin2’s phosphorylation status affected the SMN complex’s ability to condense in the nucleus. When compared to the GFP-Vector control (1.19), the overexpression of the GFP-Gemin2 wild-type protein and the corresponding phospho-mutants (1.08) had no discernible impact (Figure 5A,B). This can be explained on the one hand by the fact that the phosphorylation state of Gemin2 is not the sole factor for the number of Cajal bodies and on the other hand by the fact that in the overexpressing Gemin2 model systems, the endogenous Gemin2 is still present and performs this function. However, HEK293T cells treated with p70S6K siRNA (0.75) revealed a significant decrease in the number of CBs per cell compared to HEK293T untreated control cells (1.43) or cells transfected with non-target control siRNA (1.31) (Figure 5C,D). It is interesting to note that the SMN complex fails to condense in CBs in spinal muscle atrophy (SMA) patients [38,39]. As the siRNA-mediated knockdown of p70S6K in cells has a strong impact on the condensation of the nuclear SMN complex in vivo, this identifies p70S6K as a new putative target for the treatment of SMA patients. As mentioned above, earlier studies have linked Gemin2’s crucial role in snRNP biogenesis through regulating the binding of the Sm protein pentamer independent from pICln and thus increasing RNA binding specificity. In line with these results, a mutation in the SMN protein aborting the binding towards Gemin2, which is known to be SMA-causing, links the so-far missing mechanism of the Gemin2-mediated Sm pentamer recruitment to SMA [18]. Still, it is crucial that the role of pICln be discussed, as it is clear that it builds an RNA-free intermediate, the 6S complex, containing the Sm proteins SmD1, D2, E, F, and G. To form a functional UsnRNP, the two Sm proteins B and D3 must replace pICln. This replacement is catalyzed via phosphorylation of an mTOR pathway kinase, ULK1, which enables the transfer of the Sm proteins onto the SMN complex [28,36,37]. The new mechanism of Gemin2 phosphorylation caused by p70S6K identified here may help to explain the regulation of Sm protein transfer within UsnRNP assembly. As the phospho-status of Gemin2, influenced by the p70S6 kinase from the mTOR pathway, influences the nuclear SMN condensation in vivo, our results strengthen the assumption that Gemin2 is acting as a bridging factor between the SMN complex and the 6S complex. Thus, regulation of Gemin2 activity could have a direct influence on the formation of core UsnRNPs by mediating the transfer of Sm proteins onto the UsnRNA [18]. We hypothesize a strongly phospho-dependent mechanism of Sm protein transfer onto the SMN complex: on the one hand regulated by ULK1 via pICln [28,36] and on the other hand regulated transfer by p70S6K via Gemin2, receiving the Sm proteins from pICln. As Gemin2 conformation was shown to be open while the protein wraps around the Sm proteins [18], this conformation may be forced by phosphorylation mechanisms. Intensive work will be necessary to understand in more detail the molecular impact of p70S6K within UsnRNP biogenesis and the regulation of Gemin2 activity in this context.

## 4. Materials and Methods

### 4.1. Antibodies

The following primary antibodies were used for immunoblotting and immunofluorescence: α-coilin (Invitrogen, Carlsbad, CA, USA, #PA5-29531, rabbit), α-Gemin2 (Santa Cruz, CA, USA, #sc-166162, mouse), α-Gemin3 (Santa Cruz, CA, USA, #sc-374373, mouse), α-Gemin4 (Santa Cruz, CA, USA, #sc-365424, mouse), α-Gemin5 (Santa Cruz, CA, USA, #sc-136200, mouse), α-GFP (Invitrogen, Carlsbad, CA, USA, #14-6674-82, mouse), α-p70S6K (CST, Danvers, MA, USA, #9202, rabbit) α-SMN (Merck Millipore, Billerica, MA, USA, #05-1532, mouse), and α-SmB (Y12, Novus Biologicals, Centennial, CO, USA, #NB600-456, mouse). The detection of proteins after transfer onto PVDF membranes occurred with the following fluorescent secondary antibodies: IRDye 680LT goat α-rabbit and IRDye 800CW donkey α-mouse. For the detection of proteins in vivo via immunofluorescence, the following secondary antibodies were used: Goat anti-Mouse IgG (H + L) Cross-Adsorbed Secondary Antibody, Alexa Fluor 488 (Invitrogen, Carlsbad, CA, USA, #A11001), Donkey anti-Rabbit IgG (H + L) Highly Cross-Adsorbed Secondary Antibody, Alexa Fluor 647 (Invitrogen, Carlsbad, CA, USA, #A31573), and Alexa Fluor 568 donkey anti-mouse (Invitrogen, Carlsbad, CA, USA, #A10037).

### 4.2. Plasmids and Proteins

Plasmids containing SMN wt, and Gemin2 wt in pGEX-6P-1 Vector for protein expression were synthesized with GeneArt (Thermo Fisher Scientific, Waltham, MA, USA). For all cloning procedures, the Q5 High-Fidelity 2× Master Mix (New ENGLAND BioLabs, Frankfurt, Germany, #M0494S) and KLD Enzyme Mix (New ENGLAND BioLabs, Frankfurt, Germany, #M0554S) were used.

For cloning of the constructs in the GST-tagged pGEX-6P-1 protein expression vector, the following primers were used:

pGEX-6P-1-SMN wt, 5′-CTCGAGATGGCGATGAGCAGCGG-3′

5′-GCCCTTTTAATTTAAGGAATGTGAGCACC-3′

pGEX-6P-1-Gemin2 wt, 5′-GAATTCATGCGCCGAGCGGAAC-3′

5′-CTCGAGTCAAGATGGCTCATCAGCTAAA-3′

For cloning the Gemin2 phosphorylation mutants the following primers were used:

Gemin2 A81, 5′-GCAGTGAATATTTCTCTTTCAGGATGCCAAC-3′

5′-TTGCTTCCTTTTCAACTTCTTTGGGTC-3′

Gemin2 D81, 5′-GACGTGAATATTTCTCTTTCAGGATGCCAAC-3′

5′-TTGCTTCCTTTTCAACTTCTTTGGGTC-3′

Gemin2 A166, 5′-GCACCTGGAATAGATTATGTACAAATTGGTTTTCC-3′

5′- TTCATTTGTGGCTGGTCCAACAG-3′

Gemin2 D166, 5′-GACCCTGGAATAGATTATGTACAAATTGGTTTTCC-3′

5′-TTCATTTGTGGCTGGTCCAACAG-3′

pcDNA-FRT-TO-GFP is the vector used for the generation of inducible Flp-In T-REx 293 cells. For cloning of GFP-SMN wt, GFP-Gemin2 wt, GFP-Gemin2 A81, GFP-Gemin2 A166, GFP-Gemin2 D81, and GFP-Gemin2 D166 cell lines restriction enzymes were used with a following ligation with T4 ligase (Thermo Fisher, Waltham, MA, USA #EL0011).

### 4.3. Cell Lines and Cell Culture

Generation of inducible Flp-In T-REx 293 cell system expressing GFP, GFP-SMN wt, GFP-Gemin2 wt, GFP-Gemin2 S81A, GFP-Gemin2 S166A, GFP-Gemin2 S81D, and GFP-Gemin2 S166D were carried out according to the manufacturer’s instructions (Invitrogen, Thermo Fisher Scientific, #R78007). The generation of pcDNA5-FRT-TO-eGFP plasmid has been described previously [40]. Cells were selected with 200 µg/mL Hygromycin B Gold (Invivogen, #ant-hg-1) and 5 µg/mL Blasticidin (Invivogen, #ant-bl-05). Protein expression of Flp-In T-REx 293 cell lines was induced with 0.1 μg/mL Doxycycline (Clontech, Mountain View, CA, USA, #564-25-0) for 24 h. For p70S6K knockdown, HEK293T cells were transfected with 50 nM p70S6K siRNA (Dharmacon, #L-003616-00-0020) or SMARTpool non-targeting control (ON-TARGETplus, Dharmacon, Lafayette, CO, USA, #D-001810-10-20) for 72 h using DharmaFECT1 (Dharmacon, Lafayette, CO, USA, #T-2001-02). All cell lines were cultured at 37 °C in DMEM high glucose media (Gibco, Thermo Fisher Scientific, Waltham, MA, USA, #41965062) supplemented with 10% (*v*/*v*) FCS (Sigma Aldrich, St. Louis, MO, USA, #F9665), 100 U/mL Penicillin and 100 μg/mL Streptomycin (Gibco, Thermo Fisher Scientific, Waltham, MA, USA, # 15140122) in a 5% CO_2_ humidified atmosphere. For transfection of siRNA, opti-MEM (Gibco, Thermo Fisher Scientific, Waltham, MA, USA, #31985062) was used as a serum-reduced transfection medium.

### 4.4. Protein Expression and Purification

All proteins were expressed in BL21 DE3 Arctic express competent *E. coli* (Agilent Santa Clara, Santa Clara, CA, USA, #230192). Bacterial lysis was carried out in a buffer containing 50 mM Tris/HCl pH 7.5, 5 mM EDTA, 5 mM EGTA, 0.01% (*v*/*v*) Igepal, EDTA-free protease inhibitor cocktail (cOmplete, Roche, Switzerland, #04693132001), 50 mg/mL Lysozyme (Serva, Germany, #12650-88-3) and via sonication. After centrifugation at 15,000 rpm for 1 h, the lysate of GST-tagged proteins was incubated with pre-washed glutathione Sepharose 4B (Cytivia, Malborough, MA, USA, #17-0756-01) for 2 h at 4 °C and afterward washed 3 times with lysis buffer. For pulldown assays, recombinant proteins were incubated 2 h at 4 °C with HEK293T S100 extract and subsequently washed 3 times with lysis buffer. As a positive control, 25 µg of total protein was loaded; as a negative control, a pulldown assay with the GST protein was performed. For further analysis of the proteins, a Tris/Glycine SDS-PAGE and Western blotting with specific antibodies were carried out. As a loading control after immunoblotting, the whole membrane was stained with amido black staining (40% Methanol (*v*/*v*), 10% Acetic acid (*v*/*v*), 0.1% Amido black 10B (*w*/*v*)).

### 4.5. Cytoplasm Extraction (S100) and Size Exclusion Chromatography

After harvesting, HEK293T cells were incubated in Roeder A buffer in 3 times the volume of the weight of the cells for 10 min at room temperature. Then cells were dounced 10 times, and the lysate was adjusted to 150 mM NaCl. After centrifugation at 12,000 rpm for 30 min the supernatants (S100 extracts) were filtrated with Millex-HA, 0.45 µm filter unit (Merck Millipore, Billerica, MA, USA #HAWP04700) and applied to a Superose6 increase 10/300 GL column (GE Healthcare, Malborough, MA, USA, #GE29-0915-96). A 1 mL quantity as a fraction volume was loaded, and each 0.5 mL fraction was collected in running buffer (150 mM NaCl, 50 mM Tris/HCl pH 7.5) and analyzed via immunoblotting. The columns were calibrated with thyroglobulin (669 kDa), ferritin (440 kDa), catalase (232 kDa), aldolase (158 kDa), albumin (67 kDa), ovalbumin (43 kDa), and RNase (14 kDa) (GE Healthcare, Malborough, MA, USA).

### 4.6. Immunoblotting and Immunopurification

Protein amounts of cleared S100 cytoplasm extracts were measured via the Bradford method. Samples were separated via Tris/Glycine SDS gel electrophoresis and transferred onto PVDF membranes (Immobilon-FL, Merck Millipore, Billerica, MA, USA #IPFL00010). For the immunoblot analysis, the membranes were incubated in protein-specific primary antibodies and signals were detected with fluorescent secondary antibodies and the Odyssey LI-COR Imaging System. For GFP immunopurification, S100 extracts were incubated with pre-washed GFP-Trap beads (ChromoTek, Planegg, Germany, #gta-20) for at least 2 h at 4 °C while rotating. Immunopurified proteins were washed 3 times with Dulbecco’s phosphate-buffered saline-DPBS (Gibco, Thermo Fisher Scientific, Waltham, MA, USA, #14190144). The elution of proteins occurred in sample buffer [375 mM Tris pH 7.5; 25.8% (*w*/*v*) glycerol; 12.3% (*w*/*v*) SDS; 0.06% (*w*/*v*) Bromophenol blue; 6% (*v*/*v*) β-mercaptoethanol; pH 6.8] and analysis of samples was carried out via immunoblotting. For endogenous immunopurification, 20 µL of a 1:1 ratio mix of protein G (Cytivia, Malborough, MA, USA). #17061801): protein A (Cytivia, Malborough, MA, USA). #17127901) beads per sample was pre-incubated with 1 µg of the indicated antibody for 2 h at 4 °C. Afterward, the antibody-coated beads were incubated with HEK293T S100 extract for 2 h at 4 °C while rotating for immunopurification. After washing the samples 3 times with washing buffer (50 mM Tris pH 7.5, 150 mM NaCl, 1 mM EDTA, 1 mM EGTA, 0.01% Igepal and protease inhibitor cocktail (cOmplete, Roche, Switzerland, #04693132001)), immunoblot analysis was performed. 

### 4.7. Immunofluorescence Microscopy

Each of the 1 × 10^5^ cells of HEK293T and Flp-In T-REx 293 cell lines were seeded in DMEM high glucose media (Gibco, Thermo Fisher Scientific, #41965062) with 10% FCS (Sigma Aldrich, #F9665), Penicillin and Streptomycin (Gibco, Thermo Fisher Scientific, # 15140122) on coverslips one-day prior staining. After adhering to the cells for one night, the cells were washed once with DPBS and thus fixed with 4% paraformaldehyde for 10 min at RT. For permeabilization, cells were incubated with 0.2% Triton X-100/PBS for 10 min and blocked with 5% BSA for 30 min. Proteins were detected by incubating the following primary antibodies for 2 h: anti-Gemin2 (1:500), anti-p70S6K (1:500), anti-GFP (1:500), anti-SMN clone 2B1 (1:500), and anti-coilin antibody (1:500). As a secondary antibody, Alexa Fluor 488 (1:200; shown in magenta), Alexa Fluor 568 (1:200; shown in magenta), and Alexa Fluor 647 (1:200; shown in cyan) were used. Microscopic analysis of the antibody staining was performed with an Axio Observer microscope from ZEISS (Jena, Germany) with an ApoTome.2 and a 40× oil immersion objective. 

### 4.8. Immunofluorescence Quantification with Fiji

For the quantification of the number of Cajal bodies per cell, a macro was written in the software program Fiji, which measured the number of SMN and coilin double-positive dots within the nucleus. The macro recognized the DAPI staining, which is distributed all over the nucleus, as a region of interest (ROI). The number of either SMN or coilin dots was measured separately within the ROI. Only the ones that were SMN- and coilin-positive were counted as Cajal bodies. For every condition, 1000 cells were analyzed. The diagram and the calculation of the standard deviation as well as the statistics were made in Prism. To test the significance of the values, the data sets were analyzed using an unpaired *t*-test; the samples which were significantly different from each other had a **** *p* < 0.001. The box in the boxplot diagram represents 5–95% of the data, and outliers are shown.

### 4.9. In Vitro Phosphorylation

GST-SMN, and GST-Gemin2 wt, GST-Gemin2 S81A, GST-Gemin2 S166A, GST-Gemin2 S81D, GST-Gemin2 S166D, and GST were purified from BL21 DE3 *E. coli* (Agilent, Santa Clara, CA, USA, #230192). Recombinant active human His-p70S6K purified from Sf21 cells (Sigma Aldrich, St. Louis, MO, USA, #14-486-M) and the substrates in appropriate amounts were incubated in 2 µM ATP, 10 µCi [32P]-ATP (Hartmann Analytic, Braunschweig, Germany, #SRP-301), 2.5 mM Tris/HCl pH 7.5, 5 µM EGTA, 50 µM DTT, and 3.75 mM Mg(CH_3_COO)_2_ for 45 min at 30 °C. After terminating the reaction by adding sample buffer, samples were subjected to SDS-PAGE, and a coomassie staining following auto-radiographic analysis with Amersham Hyperfilm MP (Cytivia, #28906844) was performed.

## Figures and Tables

**Figure 1 ijms-24-15552-f001:**
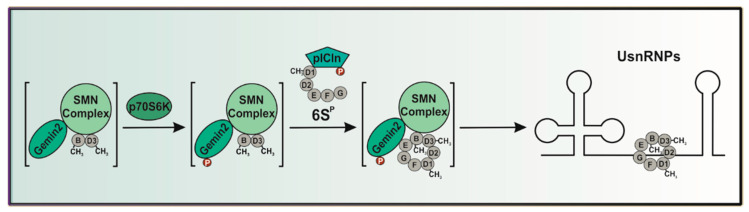
p70S6 kinase influences UsnRNP biogenesis via binding and phosphorylation of Gemin2, a core component of the human SMN complex.

**Figure 2 ijms-24-15552-f002:**
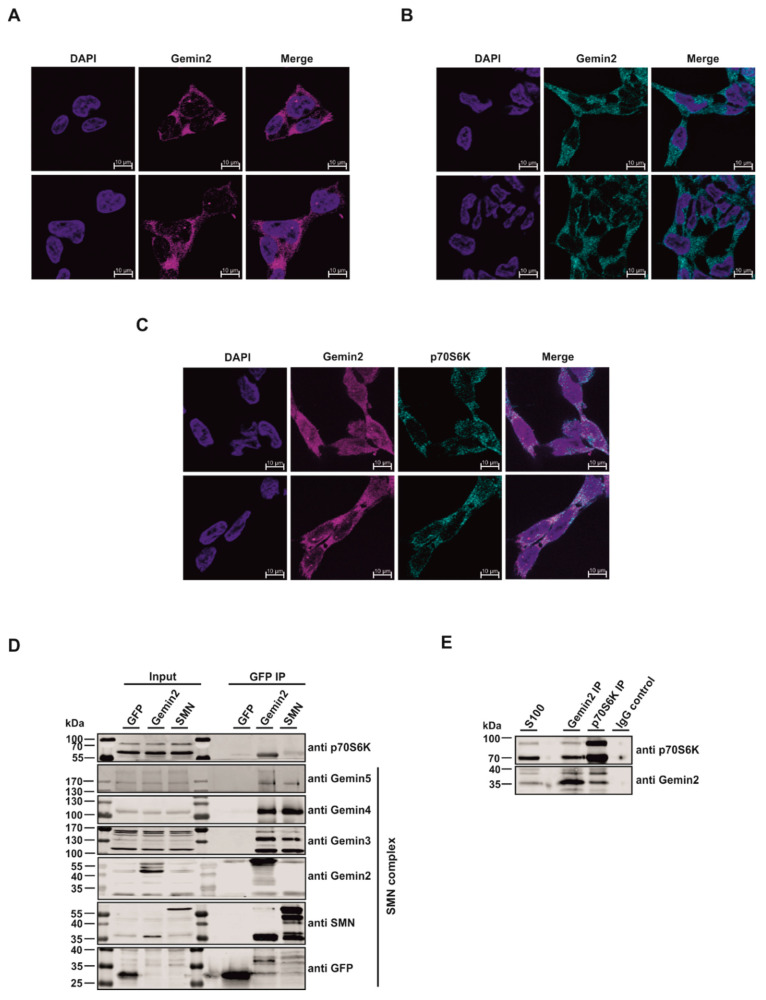
The kinase p70S6 interacts with the core components of the SMN complex in cells. (**A**) Gemin2 localizes in the cytoplasm and Cajal bodies in HEK293T wt cells. All cells were fixed with 4% paraformaldehyde and permeabilized with Triton X-100 to visualize Gemin2 (magenta). DAPI (blue) was used as a DNA marker, with scale bars of 10 µm. (**B**) The staining procedure for HEK293T cells was executed as described in (**A**), to visualize p70S6K (cyan) via specific antibody. The DNA was stained with DAPI (blue), with scale bars of 10 µm. (**C**) Gemin2 (magenta) co-localizes with p70S6K (cyan) predominantly in the cytoplasm. HEK293T cells were treated as described in (**A**). DAPI (blue) was used as a DNA marker, with scale bars of 10 µm. (**D**) Immunopurification (IP) of GFP Vector control (GFP), GFP-Gemin2 wt, and GFP-SMN wt overexpressing cells. Expression of GFP proteins was induced with 0.1 µg/mL doxycycline for 24 h. After cell lysis via douncing, GFP-IP was performed and analyzed with Tris/Glycine-SDS-PAGE and Western blotting, using antibodies against p70S6K, GFP, and core components of the SMN complex. As an input, 25 µg of total protein was loaded. The p70S6K did bind more efficiently to GFP-Gemin2 wt than to GFP-SMN wt. (**E**) Endogenous immunopurification studies revealed that Gemin2 and p70S6K interact in vivo. S100 extract of HEK293 cells was used for performing an endogenous IP experiment with each 1 µg antibody and 1 mg of total protein lysate. Afterward, the IP samples and an input sample were subjected to Tris/Glycine-SDS-PAGE followed by Western blot analysis.

**Figure 3 ijms-24-15552-f003:**
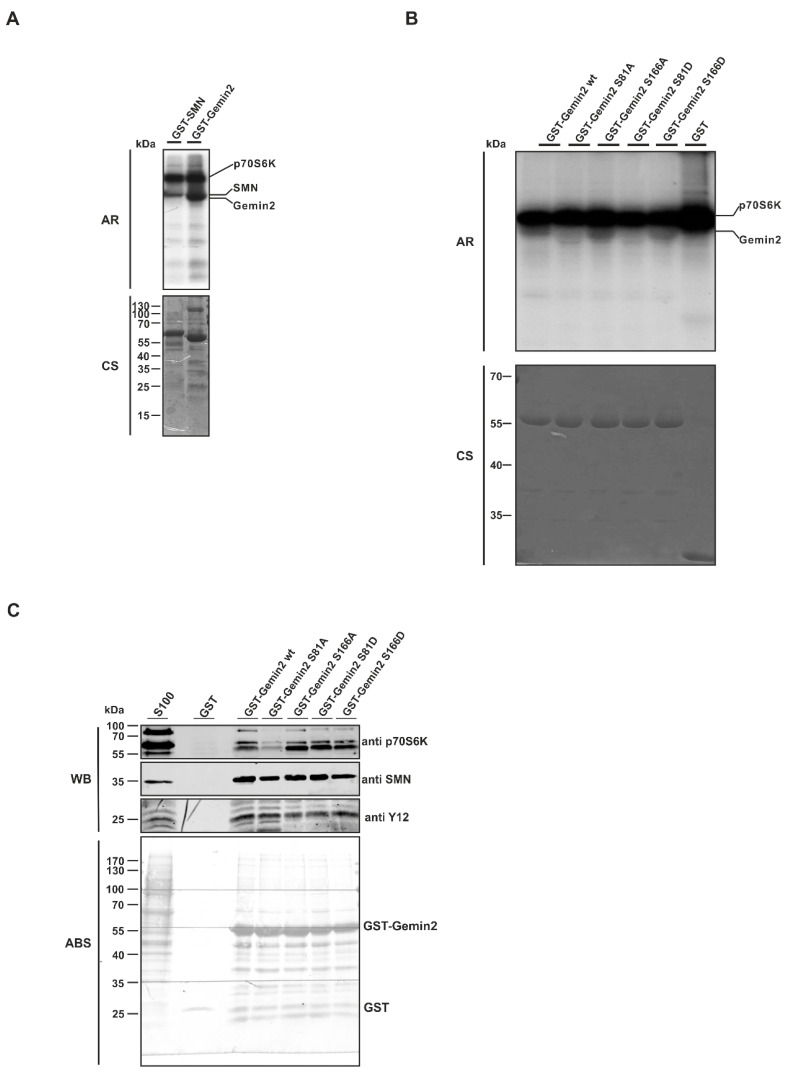
The p70S6 kinase phosphorylates Gemin2 at specific residues. (**A**) The p70S6K phosphorylates components of the SMN complex. In vitro kinase assay using recombinant active His-tagged p70S6K expressed in Sf21 insect cells and GST-SMN and GST-Gemin2 purified from *E. coli* as substrate proteins were incubated with 10 µCi [32P]-ATP for 45 min at 30 °C. Samples were separated via Tris/Glycine-SDS-PAGE and after coomassie staining (CS) analyzed by autoradiography (AR). (**B**) The p70S6K phosphorylates Gemin2 at S81. Kinase assay using His-tagged p70S6K with the substrates GST-Gemin2 wt, GST-Gemin2 SA81, GST-Gemin2 SA166, GST-Gemin2 SD81, and GST-Gemin2 SD166 as recombinant proteins purified from *E. coli* was performed as described in (**A**). (**C**) Pulldown assays using recombinant GST-Gemin2 wild type (wt), GST-Gemin2 SA81, GST-Gemin2 SA166, GST-Gemin2 SD81 and GST-Gemin2 SD166 and GST purified from *E. coli* were executed in HEK293T cytoplasmic extracts for 2 h at 4 °C. Afterward, samples were analyzed via Tris/Glycine-SDS-PAGE and Western blotting (WB), using antibodies against p70S6 kinase, SMN and SmB/B’ (Y12) and amido black staining (ABS) of the whole membrane was used as a loading control for the recombinant proteins.

**Figure 4 ijms-24-15552-f004:**
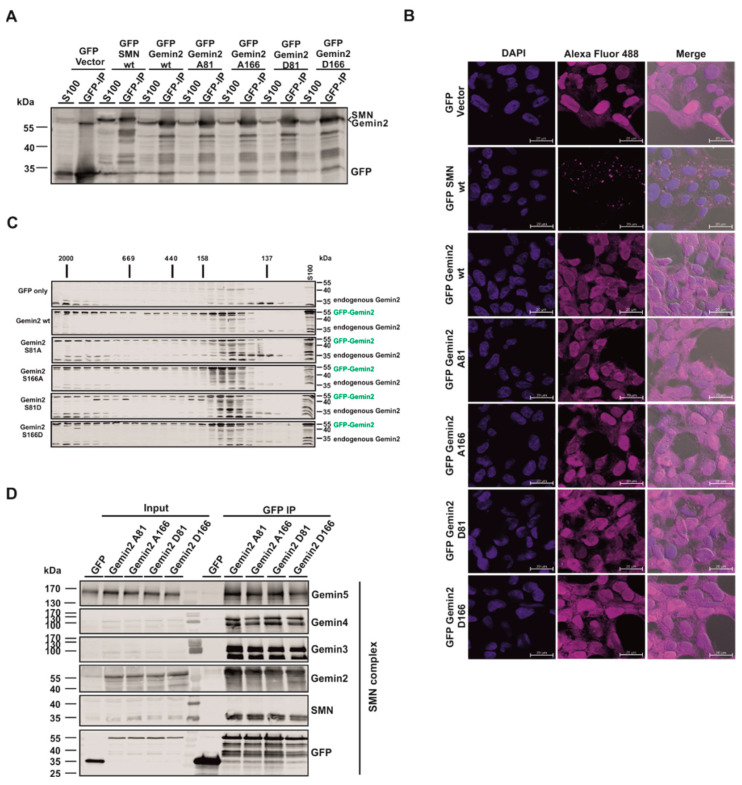
The phosphorylation status of Gemin2 does not influence the binding of the SMN complex components. *(***A**) Expression of newly generated cell lines was tested using Western blot analysis of HEK293T S100 extracts and GFP-immunopurification. IP was performed with S100 extract from GFP-Vector control, GFP-SMN wt, GFP-Gemin2 wt, GFP-Gemin2 A81, GFP-Gemin2 A166, GFP-Gemin2 D81, and GFP-Gemin2 D166 overexpressing cells. Protein expression was induced with 0.1 µg/mL doxycycline for 24 h. After cell lysis, GFP-IP was performed and together with 25 µg of total protein was analyzed via Tris/Glycine-SDS-PAGE and Western blotting, using an antibody against GFP. (**B**) Immunofluorescence studies of GFP-Vector, GFP-SMN wt, GFP-Gemin2 wt, GFP-Gemin2 A81, GFP-Gemin2 A166, GFP-Gemin2 D81, and GFP-Gemin2 D166 in overexpressing cells. Protein expression was induced as described in (**A**). Cells were fixed with 4% PFA and permeabilized with Triton X-100 to visualize the GFP overexpressed proteins (magenta). The DNA was stained with DAPI (blue), with scale bars of 20 µm. (**C**) S100 extracts of inducible Flp-In T-REx 293 cells overexpressing GFP-Vector control (GFP), GFP-Gemin2 wt, GFP-Gemin2 A81, GFP-Gemin2 A166, GFP-Gemin2 D81, and GFP-Gemin2 D166, generated by douncing, were applied to a Superose 6 column. Afterward, fractions were analyzed via Tris/Glycine-SDS-PAGE and immunoblotting using antibodies against Gemin2. Each of the overexpressed GFP and the GFP-Gemin2 variants (green) as well as the endogenous Gemin2 (black) were visualized via this analysis. (**D**) IP of GFP, GFP-Gemin2 A81, GFP-Gemin2 A166, GFP-Gemin2 D81, and GFP-Gemin2 D166 overexpressing cells. Expression of GFP proteins was induced as described in (**A**). After cell lysis, GFP-IP was performed with S100 extract and analyzed via Tris/Glycine-SDS-PAGE and Western blotting, using antibodies against GFP and core components of the SMN complex.

**Figure 5 ijms-24-15552-f005:**
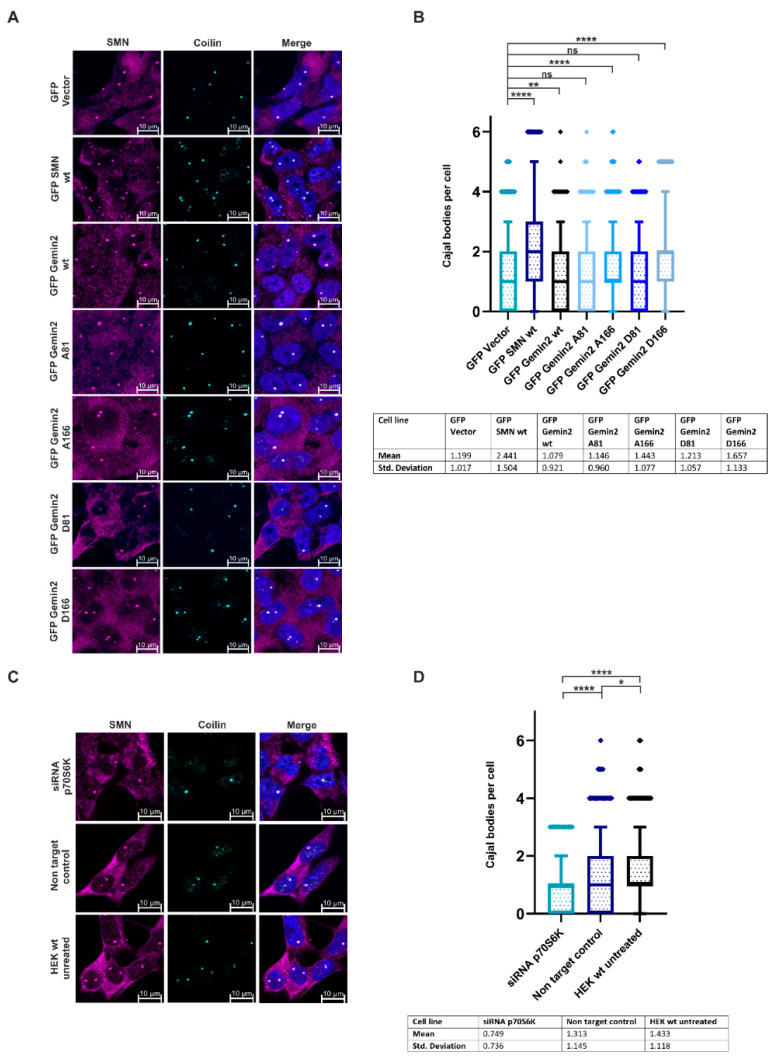
Gemin2 phospho-status and p70S6 kinase influence UsnRNP biogenesis in vivo. (**A**) Quantification of Cajal bodies in GFP-Vector control, GFP-SMN wt, GFP-Gemin2 wt, GFP-Gemin2 A81, GFP-Gemin2 A166, GFP-Gemin2 D81, and GFP-Gemin2 D166 overexpressing cells. Protein expression was induced with 0.1 µg/mL doxycycline for 24 h. Cells were fixed with 4% PFA and permeabilized with Triton X-100 to visualize SMN (magenta) and coilin (cyan). DAPI (blue) was used as a DNA marker, with scale bars of 10 µm. (**B**) Overexpression of GFP-SMN as well as GFP-Gemin2 A166 and GFP-Gemin2 D166 in Flp-In T-Rex cells caused an increase in the number of Cajal bodies compared to the GFP-Vector control and GFP-Gemin2 wt cells. The box in the boxplot diagram represents 5–95% of the data; outliers are shown as stacked rectangles. The *p*-value, calculated with Prism using an unpaired *t*-test, was **** *p* < 0.0001. (**C**) HEK293T cells were treated with 50 nM p70S6K siRNA or non-targeting control for 72 h. Untreated HEK293T cells, as well as the transfected ones, were fixed with 4% PFA, and Cajal bodies were visualized with antibody staining against coilin (cyan) and SMN (magenta). The DNA was stained with DAPI (blue), with scale bars of 10 µm. (**D**) Decrease of endogenous p70S6K resulted in a dramatic reduction in the number of Cajal bodies compared to untreated cells and non-target control. Statistics and presentation of the data were performed as described in (**B**). * for *p* < 0.05, ** for *p* < 0.01, and ns means not significant.

## Data Availability

All data and material are available upon request to christoph.peter@uni-duesseldorf.de.

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
