# Peer review of "The Impact of p70S6 Kinase-Dependent Phosphorylation of Gemin2 in UsnRNP Biogenesis"

_ijms, 2023, doi:10.3390/ijms242115552_

Round 1

Reviewer 1 Report

The review by Esser et al reports new data on the phosphorylation status of Gemin2, a core component of the survival motor neuron (SMN) complex, and its effects on SMN complex condensation in Cajal bodies. The authors identify p70S6K kinase as a putative kinase responsible for phosphorylation of Gemin2 and confirm serine 81 as one of the phosphorylation sites. In general, the manuscript is based on solid experiments and the results support well the main conclusions of the manuscript. I have only a few relatively minor comments.

 1.      The introduction would benefit from a figure schematically depicting the cellular processes in which SMN is involved and also highlighting the role of Gemin2 phosphorylation along the pathway, thereby expressing the main hypothesis of the article.

2.      The part of the introduction that presents the conflicting/contradictory results of previously published research is not clear – lines 53-58l - with whom do the authors agree?

3.      Results – the nomenclature – I would suggest that the authors always write out the name of the fusion in full, otherwise it is confusing. E.g. line 83: “GFP-SMN and GFP-Gemin2 wild type” instead of “GFP-SMN and -Gemin2 wild type”.

4.      Fig. 1A: Cajal bodies are not too visible, consider a larger and/or better resolution image.

5.      Fig.1, 2 and 3 and/or results: it would be good to specify the molecular weights of analyzed proteins.

6.      Line 109 – to determine whether, or, to assess if….

7.      Fig. 2B – the exposure is too large to quantify the reduction in signal. In addition, the bands/signals of p70S6K and Gemin2 are not sufficiently resolved. It is also surprising that the signal belonging to the kinase is so strong in the control well (last well, GST only). I would suggest the authors to repeat the experiment with a longer run time of the gel electrophoresis and shorter exposition time for autoradiogram development (as in Fig. 2A).

8.      Fig. 2C: The authors might consider quantifying normalized band intensities and adding graphical representation of intensities to the Western blot.

9.      Results, section 2.4: A better presentation of the results is needed because this part reveals data that are not very well connected to the previous parts. A reduction in CBs is observed in the S166 mutants, whereas no effect is observed in the S81 variants. Furthermore, knocking down p70S6K decreases the number of CBs, which is not in agreement with the fact that the S81 variants have no effect on the number of CB.

10.   Line 209: What is “slide” reduction?

11.   Line 214: I would weaken this conclusion because other effects could come into play and not just phosphorylation.

12.   Discussion: Line 168, serine 81, not just serine.

13.   Line 279: “inducible overexpressing” – consider rewording,

14.   Some of the figures are not mentioned in the discussion, please check.

There are some syntactical and punctuation errors, so I recommend checking the manuscript carefully.

Reviewer 2 Report

In the present manuscript, “ The impact of p70S6 kinase-depended phosphorylation of Gemin2 in UsnRNP biogenesis” the authors have provided strong experimental evidence to prove the significant role of Gemin2 in uridine-rich short nuclear RNA (UsnRNA) biosynthesis. The authors have experimentally shown the interactions of p70S6 kinase interactions with the SMN complex using immunoprecipitation of indicated proteins. The authors’ work has identified the phosphorylation site of Gemin2 as S28. Moreover, this manuscript provided additional mechanistic evidence to strengthen the assumptions that Gemin2 is a bridging factor between the 6S complex and the SMN complex involved in transferring the Sm proteins from the 6S 293 complex onto the snRNA.

This manuscript is complete and should be accepted in present form for publication in IJMS. 

Author Response

We totally agree with the reviewer.